# Comparative Analysis of the Mitochondrial Genome of Eggplant (*Solanum melongena* L.) to Identify Cytoplasmic Male Sterility Candidate Genes

**DOI:** 10.3390/ijms25179743

**Published:** 2024-09-09

**Authors:** Wentao Deng, Guiyun Gan, Weiliu Li, Chuying Yu, Yaqin Jiang, Die Li, Qihong Yang, Wenjia Li, Peng Wang, Yikui Wang

**Affiliations:** 1Vegetable Research Institute, Guangxi Academy of Agricultural Sciences, Nanning 530007, China; xt1128328@foxmail.com (W.D.); ggyun19@163.com (G.G.); liweiliu@gxaas.net (W.L.); yuchuying@126.com (C.Y.); jiangyaqin@gxaas.net (Y.J.); lid10949@163.com (D.L.); yangqihong@gxaas.net (Q.Y.); lwj3386@gxaas.net (W.L.); 2Agricultural College, Guangxi University, Nanning 530004, China

**Keywords:** CMS-associated gene, comparative genomics, gene mining, eggplant (*Solanum melongena* L.) molecular marker

## Abstract

Cytoplasmic male sterility (CMS) is important for commercial hybrid seed production. However, it is still not used in eggplant (*Solanum melongena* L.), and corresponding regulatory genes and mechanisms of action have not been reported. We report CMS line 327A, which was derived from the hybridization between cultivated and wild eggplants. By looking at different stages of anther development under a microscope, we saw that the 327A anther’s tapetum layer vacuolized during meiosis, which caused abortion. To investigate the 327A CMS regulatory genes, the mitochondrial genomes of 327A and its maintainer line 327B were assembled de novo. It was found that 15 unique ORFs (Open Reading Frame) were identified in 327A. RT-PCR and RT-QPCAR tests confirmed that *orf312a* and *orf172a*, 327A-specific ORFs with a transmembrane domain, were strongly expressed in sterile anthers of 327A. In addition, *orf312a* has a chimeric structure with the ribosomal protein subunit *rpl16*. Therefore, *orf312a* and *orf172a* can be considered strong candidate genes for CMS. Concurrently, we analyzed the characteristics of CMS to develop a functional molecular marker, CMS312, targeting a future theoretical basis for eggplant CMS three-line molecular breeding.

## 1. Introduction

Eggplant is an often-self-pollinated plant with obvious heterosis, especially in yield and maturity, and exhibits better-parent heterosis [1]. At present, approximately 80% of the main cultivars of eggplant are hybrid seeds [2], but conventional hybrid seed production has the problems of high labor costs and difficulty guaranteeing seed purity. CMS is a maternally inherited trait that can reliably produce sterile plants. When used as a parent in crossbreeding, it can effectively avoid self-pollination and improve breeding efficiency [3].

The matrilineal inheritance phenomenon of CMS is thought to originate from the mitochondria. And the size of the mitochondrial genome varies widely between species, with a range of 68–11,319 kb [4,5]. The eggplant’s mitochondrial genome is around 500 kb, as discovered by our team’s previous study [6]. Compared to animals and fungi, the mitochondrial genome of plants is especially prone to genome recombination and rearrangement, generating subgenomic forms and extensive structural variation. And a number of new open reading frames (ORFs) have been generated. These novel ORFs frequently have chimeric structures or are co-transcribed with known mitochondrial genes, which causes known mitochondrial genes to lose function [7]. They are also often associated with the respiratory chain, e.g., causing reduced ATP synthase activity, excessive ROS accumulation, or premature PCD (programmed cell death) [8].

Mitochondrial function genes that do not work correctly can affect how energy is made and transferred. This can cause the anthers to grow incorrectly and the mature pollen grains to decrease, which makes the male plant sterile. So, studying cells during the development of the anther gives us a solid base for learning how sterility genes are controlled in pollen abortion. The occurrence of various CMS lines is temporally distinct, but the majority are associated with abnormalities in the tapetum layer. Brassica napus CMS105A microspores broke down because of an abnormal tapetum in the late mononuclear stage [9]. The pepper CMS B351A was caused by the tapetum becoming hollow during meiosis, which made the pollen mother cells not form properly [10]. In eggplant, Chen Xue Ping [11] analyzed the anthers of several CMS strains and categorized them into three primary types: anther deterioration, anther petaloid, and anther apical petaloid. Abnormalities in the structure and function of the tapetum layer are the common cause of all three types.

To date, regulatory genes for CMS have been identified in about 22 species. These genes are usually independent open reading frames (ORFs) that encode short proteins similar to mitochondrial functional genes or are associated with the structure of known mitochondrial functional genes. *Orf79* and its variant *orfH79* in CMS-BT and CMS-HT rice make short N-terminal sequences that are similar to *cox-1* [12,13]. In CMS tomatoes, *orf265* shows similar sequences to *atp8* [14]. In another case, ORF is transcribed together with mitochondrial functional genes. *Orf352* has been reported to be co-transcribed with *rpl15* in CMS-RT102 rice [15], and *orf98* has a chimeric structure with *atp4* and *cox3* in CMS-RT98 [16]. In CMS97A tropical onion, *orf725* has been reported to be co-transcribed with *cox1* [17]. *Orf606* and *atp1* were co-transcribed in the cotton CMS-D2 line [18]. Also, CMS regulatory genes were found in Solanaceae pepper plants: *orf300a* [19], *orf507* [20], *orf456* [21], and *orf165* [22] were all associated with aberrant expression of ATP or COX. In eggplant, *orf218* and *orf312* were identified as CMS regulation genes, which share transcripts with *atp1* [23]. So far, the mitochondrial genome of the CMS line in eggplant has not been previously documented or examined, and other sources of sterility are to be further explored.

In this study, we first completed the sequencing and assembly of the mitochondrial genomes of the CMS line 327A and its maintainer line 327B. Then, by observing anthers at different stages, we clarified the cytological mechanism of 327A abortion. Comparing mitochondrial genomics showed differences in sequences and genes between 327A and 327B. Based on this, the genes controlling cytoplasmic male sterility in eggplants were predicted and corresponding functional molecular markers were developed. This study contributes to a further understanding of the molecular mechanism of CMS327A abortion as well as the development and application of CMS lines in eggplant.

## 2. Results 

### 2.1. Sterility Characters and Cytological Observation

Comparing the mature-period flower of CMS327A with its maintenance line, 327B, showed that its anthers deteriorated at maturity (Figure 1). Then, paraffin sections were produced at distinct developmental stages to clarify the timing and process of abortion. Cytological examinations showed no significant variation in the sporulation stage between 327A and 327B (Figure 2A,G). During meiosis, the 327A sporoblast extruded, hollowing the tapetum layer (Figure 2C). In the tetrad stage, the sterile line 327A’s tapetum layer separated too early, slowing callose breakdown and microspore release (Figure 2D). The sterile line 327A’s tapetum is constricted into bands at the binucleate and mature stages. This prevented pollen development; therefore, no grains formed (Figure 2J,K). In the maintenance line, ripe pollen grains were visible, and the meiosis, tetrad, binucleate, and maturity stages were normal.

### 2.2. Assembly and Annotation of the Mitochondrial Genome

The mitochondrial genomes of sterile line 327A and maintainer line 327B were obtained by de novo assembly. Line 327A’s total length was 459,611 bp, and the GC content was 44.59%. Line 327B’s total length was 520,408 bp, and the GC content was 44.62%. After that, ORFs with more than 100 amino acids were found in the mitochondrial genome that had been put together. In 327A, 24 ncRNAs and 172 ORFs were found. In 327B, 34 ncRNAs and 194 ORFs were found. According to genomic annotation information, 42 and 49 known genes were found in 327A and 327B, respectively. In the known genes, there is correspondence, but there are some differences between the sterile line 327A and the maintainer line 327B. The differences include ATP synthase protein (*atp8-D2*), ATP synthase subunit (*atp9-D2*), cytochrome oxidase subunit (*cox3-D2*), TATC-like protein (*mttB-D2*), NADH (Nicotinamide adenine dinucleotide) dehydrogenase subunit (*nad1-D2*, *nad2-D2*, *nad7*), ribosomal protein (*rps19-D3*), and succinate dehydrogenase subunit (*sdh4-D2*). This might be associated with duplicate genes (Appendix A). Finally, DNA sequences and annotation results were used to make mitochondrial genome maps of eggplant sterile line 327A and maintainer line 327B using the software OGDRAW 1.3.1. (Max-Planck-Institut für Molekulare, Potsdam-Golm, Germany) [24] (Figure 3).

### 2.3. Mitochondrial Genome Comparison of 327A and 327B

#### 2.3.1. SNP and InDel Analysis

To determine the cause of CMS, we identified the sequence variation for known genes. A total of 288 SNPs and 86 InDel mutations existed between the two mitochondrial genomes (Appendix A). Of the 288 SNPs, there were five synonymous mutations, including *cox2*, *cob*, *rpl2*, and *nad4L*, as well as mutations in these genes that are not normally thought to alter the amino acids they code for. There are 19 non-synonymous mutations, mainly involving *sdh3*, *ccmFc*, *rpl10*, *rpl16*, *atp6*, and *rps1*. Amino acid point mutations caused by these non-synonymous mutations are potential factors for studying CMS’s molecular mechanism (Appendix A). In terms of InDel, 28 insertions and 58 deletions occurred in intergenic regions (Appendix A).

#### 2.3.2. Structural Variation Analysis

Chimeric ORFs are generated by mitochondrial recombination, which is induced by the many repetitive sequences in the mitochondrial genome. The mitochondrial genomes of 327A and 327B have many repetitive sequences, with 327B having a maximum of 25,283 bp and 327A 657,2 bp. This shift may be attributed to mitochondrial genome recombination. 327A had 873 repeats of over 30 base pairs, while 327B had 718. The most common repeats were 30-34 base pair sequences: 356 in 327A and 423 in 327B (Appendix A). To further explain the differences between the mitochondrial genomes of 327A and 327B, the structural variation of 327A was analyzed using 327B as a reference. This analysis identified 29 translocations and inversion regions, as well as 15 translocations and 5 inversion regions. Furthermore, two insertions larger than 50 base pairs, four deletions, and three composite regions were detected (Figure 4).

### 2.4. Analysis of Endemic ORF and Identification of CMS Candidate Genes

Due to the rearrangement of the mitochondrial genome, 102 new ORFs were produced in 327A (Appendix A, Appendix A). The collinear analysis of 327A and 327B revealed that out of the 62 blocks compared, 13 blocks were found to be homologous sequences, accounting for 85.2% and 75.27% in 327A and 327B, respectively (Figure 5A). A total of 38 unique ORFs were identified within 14.8% of the region where structural variants such as transitions and inversions occur in 327A (Appendix A). The 102 and 38 ORFs were analyzed to find 15 unique ORFs in the 327A mitochondrial genome rearrangement-induced structural variation. Based on previous research, the regulatory genes of CMS are chimeric structures, co-transcribed with known mitochondrial genes, or encoded peptides with transmembrane domains. *Orf321a* and *Orf172a* were selected from 15 unique ORFs as CMS candidate genes based on the above characteristics. *orf312a* and *orf561a* were chimeric structures from the 5’ and 3’ ends, respectively, in a co-transcription of 2996 bp with the shared sequence for *rpl16*. *ORF172* is a unique ORF with three transmembrane domains. To further clarify the effects of *orf312a* and *orf172a* on CMS, we confirmed the existence of the common transcript of orf561a-rpl16-orf312a by RT-PCR, and *orf172a* was amplified only at 327A (Figure 5C). In addition, *orf312a* and *orf172a* were not expressed in 327B by RT-QPCR, and *rpl16* was not expressed differently between 327A and 327B. This is consistent with the results of RT-PCR (Figure 5D). Based on these results, we suggest that *orf312a* and *orf172a* are candidate genes for cytoplasmic male sterility in 327A.

### 2.5. Developing Molecular Markers for CMS

We developed a molecular marker called CMS312 based on *orf312a* to distinguish the fertility of eggplant pollen. The results showed that this molecular marker could effectively distinguish the 17 eggplant-bred lines tested (Figure 6). DNA fragments were amplified in 327Q2, 327Q3, 327Q7, and 327Q16, and bands appeared at about 3000 bp. Compared with the actual fertility, the accuracy was 100 %. These results indicated that CMS312 markers could be used to screen CMS traits in eggplant and could be used as a source for matching the breeding of cytoplasmic male sterility triple crosses in the future.

## 3. Discussion

Cytoplasmic male sterility (CMS) is an important means to realize the advantages of crop hybridization; it saves time and effort, simplifies seed production procedures, and has high seed purity. Compared with other solanum vegetables, eggplant’s male sterility has not developed a large area of effective matching for production across three lines, and the related theoretical research is relatively lagging. Due to the unique advantages of the CMS systems, CMS lines of eggplant were successively created through distant crosses with *Solanum violaceum*, *Solanum anguivi*, and *Solanum grandifolium* [26,27,28]. However, there are few studies related to sterility-regulated genes.

Some CMS-regulated genes have been characterized as base mutations in mitochondrial functional genes and structural variants resulting in code-shifting mutations [17,18]. So, we first checked for differences in the annotated genes of the two genomes. In the annotated genes of the mitochondrial genomes of 327A and 327B, nine genes were found to be missing in the CMS line 327A compared with the maintainer line 327B. These genes are *atp8-D2*, *atp9-D2*, *cox3-D2*, *mttB-D2*, *nad1-D2*, *nad2-D2*, *nad7-D2*, *rps19-D3*, and *sdh4-D2*. Furthermore, a comparative analysis revealed 19 non-synonymous SNP mutations between 327A and 327B, none of which caused transcriptional disturbances. The InDel site is also absent in the gene’s coding region.

Based on numerous studies of CMS regulatory mechanisms, CMS regulatory genes exhibit the following characteristics: they are associated with CMS-specific ORFs; are co-transcribed with known gene structures in mitochondria; and contain one or more transmembrane domain.

Due to the rearrangement of the mitochondrial genome, the original sequence order is disrupted, leading to the creation of new ORFs unique to CMS lines. These CMS-specific ORFs are often functionally similar to known mitochondrial genes. For instance, *orf137* [29] in tomatoes is comparable to the sequence of cytochrome C subunit 1, which plays a role in cell respiration to some extent, affecting the generation and transportation of energy [30]. The more common case for CMS genes is that the endemic ORF and known genes are chimeric and co-transcribed. However, the common transcription causes dysfunction or loss of the original gene function, which leads to metabolic abnormalities and abortion. In the further screening of candidate genes, we took the above conditions into account and analyzed two CMS candidate genes: *orf312a* and *orf172a*. Both *orf172a* and *orf312a* have multiple transmembrane domains and exist in a unique region of 327A. In addition, *orf312a* and *orf561a* form a chimeric structure of about 3000 bp with the ribosomal protein subunit *rpl16* from the 3′ terminal and 5′ terminal, respectively. However, *orf561a* is not a specific ORF in 327A and does not have a transmembrane domain. At the same time, we further verified the differences in the expression levels of *orf312a* and *orf172a* between 327A and 327B by RT-PCR and RT-QPCR. In summary, we identified *orf312a* and *orf172a* as strong candidate genes for CMS327A. Finally, to efficiently utilize CMS sterile line 327A in eggplant, we developed a functional molecular marker, CMS312, based on an *orf312a*-specific sequence, that can quickly and accurately identify CMS in eggplant. Nevertheless, there are still some potential problems to be solved, for example, the direct functional validation and mechanism analysis of *orf312a*, the creation of the restorer line, and the mining of genes for fertility restoration. And these are the focus of future research.

In conclusion, we finished assembling and sequencing the complete mitochondrial genome of the CMS line 327A and its maintainer line 327B. This work provides valuable resources for gaining insight into the origin and evolution of mitochondrial genes in eggplant. Then, we identified and analyzed the candidate genes for CMS regulation, *orf312a* and *orf172a*, and developed CMS markers based on the specific sequences of *orf312a*. This discovery opens up the possibility of artificially regulating fertility and improving CMS breeding efficiency in eggplant breeding in the future.

## 4. Materials and Methods

### 4.1. Plant Materials

In this study, we used the eggplant cytoplasmic male sterile line 327A and its maintenance line 327B as test materials. 327A was derived from a cross between cultivated and wild eggplant, and the maintenance line 327B was one of its rotational crossing’s parents. (327A and 327B were manufactured in Guangxi Academy of Agricultural Sciences, Nanning, China) They were cultivated under standard growing conditions (longitude 108°, latitude 23°) at the Li Jian Experimental Base of the Guangxi Academy of Agricultural Sciences, Nanning, Guangxi, China. A total of 100 seeds were sown in seedling pots at 28 °C for germination of 327A and 327B, and when the seedlings reached 4 leaves and 1 heart, the young roots were collected and snap-frozen in liquid nitrogen to extract mitochondrial DNA.

### 4.2. Paraffin Sections and Cytological Observations

Flower buds of 327A and 327B were collected at six periods of pollen development for the preparation of paraffin sections. Microscopic observations of microspore development were made to determine the characteristics of microspore abortion in CMS line 327A and the cytological basis of its occurrence.

### 4.3. De Novo Mitochondrial Genome and Assembly

In mitochondrial genome sequencing, we use second-generation Illumina NovaSeq 6000 in combination with third-generation Nanopore sequencing. First, we extracted eggplant 327A and 327B total DNA using the E.Z.N.A.^®^ xxx DNA kit (OMEGA, Cambrige, MA, USA). Then, we performed library construction using the TruSeqTM Nano DNA Sample Prep Kit (IIIumina, San Diego, CA, USA) according to the reagent manufacturer’s instructions and quantified and recovered libraries using TBS380 Picogreen (Invitrogen, Carlsbad, CA, USA) and Certified Low Range Ultra Agarose (Bio-Rad, Hercules, CA, USA), respectively. Afterwards, cBot Truseq PE Cluster Kit v3-cBot-HS (IIIumina, San Diego, CA, USA) was used for bridge amplification and finally 2*150 bp sequencing was performed on the Lumina NovaSeq sequencing platform. Specifically, we used NovaSeq6000 to generate 8373.4 Mb and 4938.8 Mb of pure read length from 327A and 327B, respectively. The corresponding Q30 values were 94.12% and 96.51%. Subsequently, filtered read sizes of 640 Mb and 786 Mb were obtained from 327A and 327B, respectively, using Nanopore.

For genome assembly, the Illumina sequencing data were first assembled using GetOrganelle v1.7.1 (Chinese Academy of Sciences, Kunming, China) [31] and then the second-generation assembled sequences were aligned with the Nanopore third-generation data using BWA v0.7.17 (Harvard University Cambridge USA) [32] to extract the third-generation data of the target samples by software Canu V2.2 (Intramural Research Program, Bethesda, MD, USA) [33], and then the extracted third-generation data were mixed and assembled with the second-generation data for hybrid assembly(by hybridSPAdes, Algorithmic Biology Laboratory, St. Petersburg Academic University, Russian Academy of Sciences, St. Petersburg, Russia) [34]. Using the comparison software ncbi-blast-2.8.1 (National Library of Medicine, Bethesda, MD, USA) + and the threshold e value 1e-5, sequences with sufficiently high coverage depth and long assembly length were selected as candidate sequences, and the mitochondrial sequences were confirmed based on the overlap by comparing them with NT library scaffolding sequences and linkage sequences. Clean reads were then aligned to the mitochondrial genome sequence and corrected for bases using Pilon v1.23 (Cambridge, MA, USA) [35]. Finally, the start position and orientation of the mitochondrial scaffold were determined from the reference genome to obtain the final mitochondrial genome sequence by NOVOPlasty (Interuniversity Institute of Bioinformatics in Brussels, Brussels, Belgium) [36].

### 4.4. Gene Prediction and Functional Annotation

The eggplant mitochondrial genome was genetically predicted using a combination of homology-matching prediction and de novo prediction. Protein sequences from the NCBI mitochondrial reference genome (MF034194, Solanum pennellii biomaterial, and TGRC:LA0716 mitochondrion) were first quickly aligned to the sample genome using the software BLAST+ 2.8.1(National Library of Medicine, Bethesda, MD, USA), with the threshold parameter set at e-value 1 × 10^−5^. Bad alignments were filtered, redundancies were removed, and then the sample genes were manually corrected for completeness and exon/intron boundaries to obtain a conserved set of genes with high accuracy. The tRNA scan-SE v2.0.7 (Santa Cruz, CA, USA) and rRNA mmer 1.2 (DTU health tech, Copenhagen, Denmark) were used to predict the genes that code for transporter RNAs and ribosomal RNAs, respectively. Thegetorf (EMBOSS 6.6.0, http://emboss.sourceforge.net/, accessed on 16 April 2024) software was used to predict ORFs with an amino acid number of 100 or more in the eggplant mitochondrial genome. The parameters were set as in Appendix A, including minsize 300. In addition, the transmembrane structure of ORFS was analyzed using TMhmm 2.0 software (DTU health tech, Copenhagen, Denmark http://www.cbs.dtu.dk/services/TMHMM-2.0/, accessed on 18 April 2024). Finally, the gene sets were corrected and integrated to obtain the sample’s genome-coding genes.

The amino acid sequence of the predicted gene was compared to a known protein database using specific criteria (e-value ≤ 1 × 10^−5^, matching length percentage ≥ 40%). The predicted gene and its corresponding functional annotation information were combined to obtain the annotation results. To ensure biological significance, the best alignment result for each sequence was retained as the gene’s annotation. The databases used for functional annotations are NR (http://www.ncbi.nlm.nih.gov/, accessed on 21 April 2024), Swiss-Prot (http://www.ebi.ac.uk/uniprot, accessed on 21 April 2024), COG (http://eggnogdb.embl.de/, accessed on 23 April 2024) [37,38], KEGG (http://www.genome.jp/kegg/, accessed on 23 April 2024) [39], GO (http://geneontology.org/, accessed on 25 April 2024), and Geseq (https://chlorobox.mpimp-golm.mpg.de/geseq.html, accessed on 26 April 2024) [40].

### 4.5. Collinearity and Variation Analysis

First, the MUMmer v3.23 (Rockville, MD, USA) program was used to examine the collinearity of the sterile line 327A and the maintenance line 327B to determine how the genomes relate on a broader scale. The comparison of regions was done with LASTZ v1.03.54 (Miller Lab, University Park, PA, USA). Find genomic structural variation (DNA fragment deletion, insertion, duplication, inversion, and ectopic). The 327A and 327B sequences were globally compared using MUMmer v3.23 to find and filter differences and discover SNP locations. Sequences of 100 base pairs on each side of the 327A SNP site were retrieved. To confirm the SNP location, BLAT v35 software compared these extracted sequences to the assembly findings. To find 1–10 base pair InDels, LASTZ software was used to match the 327A reference sequence to the 327B mitochondrial genome sequence. The reference sequence’s InDel sites were compared to 327B using BWA software and tools, yielding trustworthy InDels after filtering.

### 4.6. Analysis of Candidate Gene Expression

Flower buds of CMS line 327A and its maintenance line 327B at different periods were used to determine the expression of candidate ORFs. We extracted the total RNA using the SV Total RNA Isolation System Kit (Promega, Madison, WI, USA) and the Plant Total RNA Extraction Kit (Shanghai Promega, Shanghai, China). For semi-quantitative RT-PCR, reverse transcription into cDNA was performed: pre-denaturation at 94 °C for 5 min, denaturation at 98 °C for 10 s, annealing at 55 °C for 15 s, extension at 72 °C for 50 s, 38 cycles, and extension at 72 °C for 5 min. PCR products were detected by 1% agarose gel electrophoresis. Our RT-QPCR was done with the TB Green TM Premix Ex Taq TM kit (Takara Bio, Beijing, China), and three technical and biological replicates were set up. The PCR reaction program was as follows: 95 °C for 30 s, 95 °C for 5 s, 60 °C for 34 s, and 40 cycles; melt curve (Appendix A lists the primers).

## Figures and Tables

**Figure 1 ijms-25-09743-f001:**
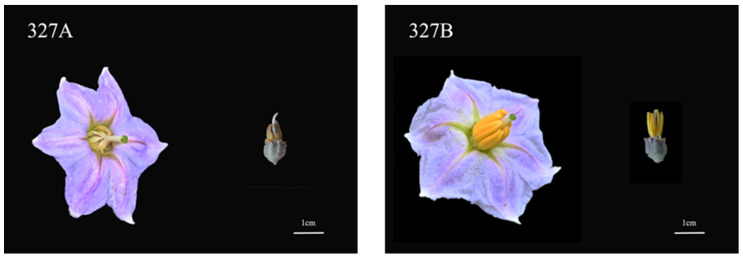
The mature flower morphology of CMS line 327A and maintainer line 327B. (The left side displays CMS mature flowers and bud morphology, while the right side shows maintainer mature flowers and bud morphology. Scale: 1 cm.)

**Figure 2 ijms-25-09743-f002:**
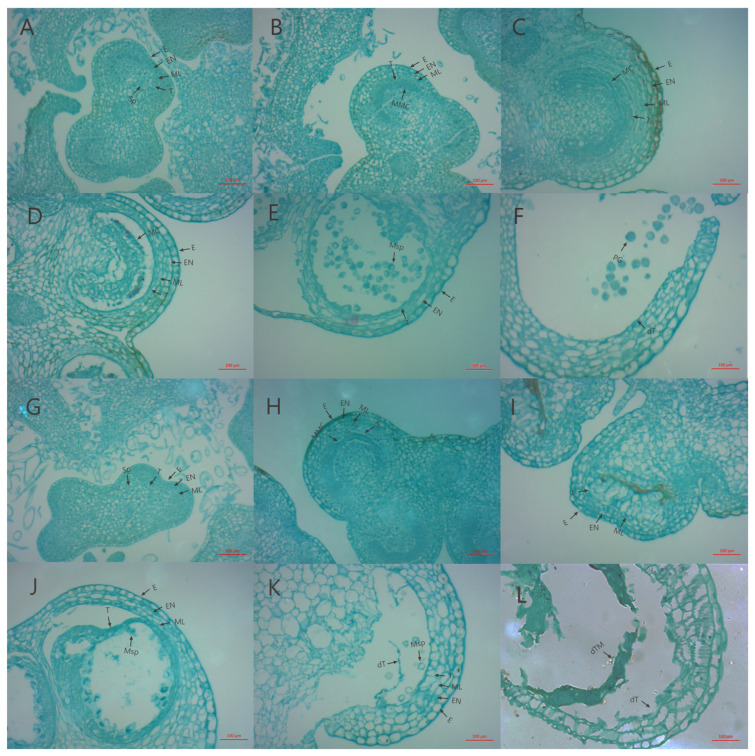
Analyzing the anther cytology of 327A and 327B at different periods. (**A**–**F**) are anther paraffin sections of maintainer line 327B at different periods; (**G**–**L**) are paraffin sections of anther of sterile line 327A at different periods. (**A**,**G**) are spore-forming cell stages; (**B**,**H**) are the microspore mother stage; (**C**,**I**) are the meiosis phase; (**D**,**J**) are the tetrad periods; (**E**,**K**) are the binuclear phase; (**F**,**L**) is the mature stage. E: epidermal layer; EN: endodermis; ML: middle layer; T: tapetum layer; Sp: spore-forming cell; MMC: pollen mother cell; Msp: microspore; PG: pollen grains; dT: apoptotic tapetum layer; dTM: apoptotic tapetum and microspore. Scale= 100 μm.)

**Figure 3 ijms-25-09743-f003:**
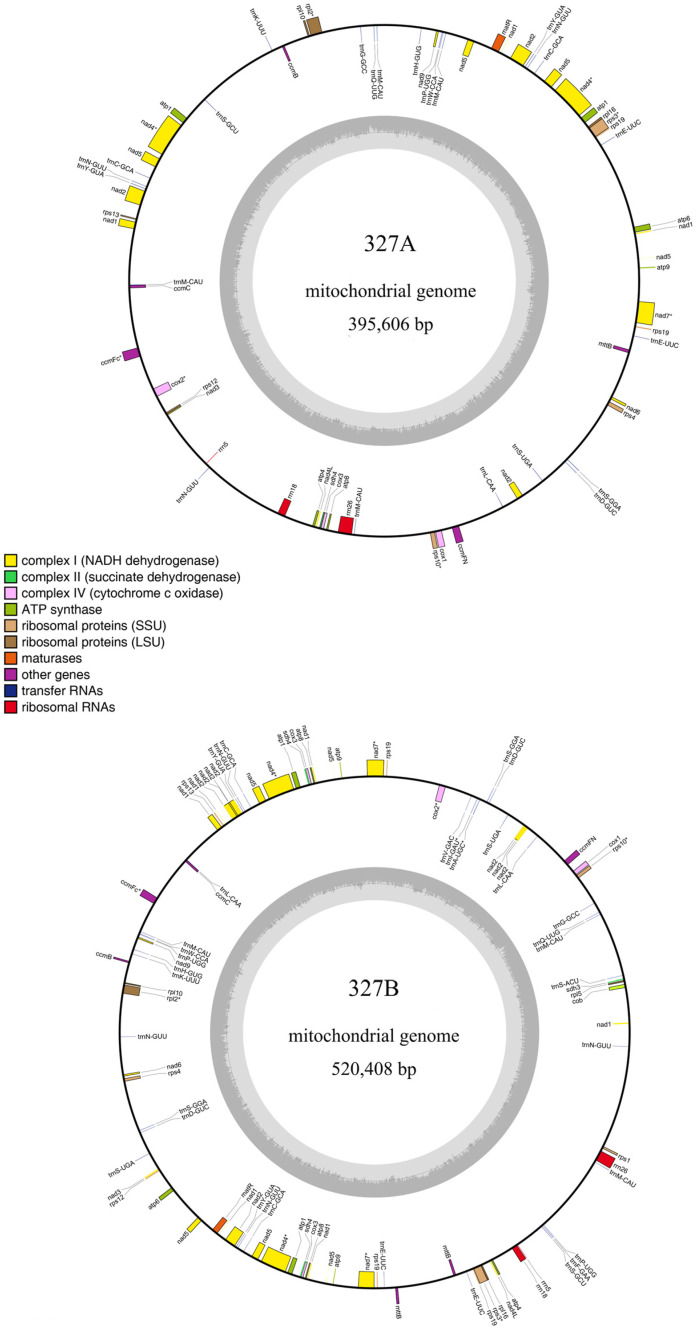
Mitochondrial assembly maps of sterile line 327A and maintainer line 327B of eggplant. (The outer circle is the location coordinates of genomic components such as genes and ncRNAs, with corresponding gene names; the inner circle is the genomic GC content; different colored blocks represent the functions of different gene products.)

**Figure 4 ijms-25-09743-f004:**
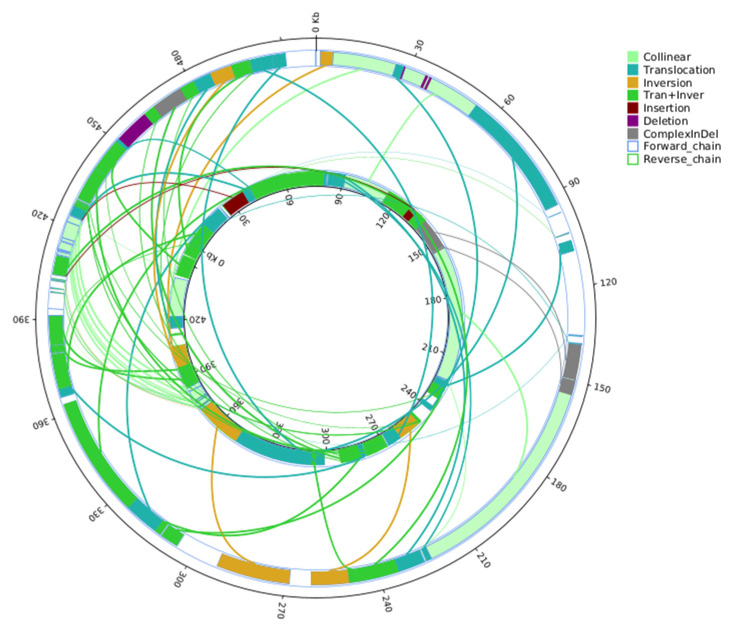
The inner circle of the mitochondrial genome structure comparison diagramis the maintainer line 327B genome, and the outer circle is the sterile line 327A genome. And this diagram plotted by SyRI (Max Planck Institute for Plant Breeding Research, Planegg-Martinsried, Germany) [25]. (Collinear: homo-linear region; Translocation: translocation region; Inversion: an inversion area; Tran+Inver: translocation and inversion region; Insertion: insertion areas of 50 bp or longer; Deletion: deletion area of length greater than or equal to 50 bp; Complex InDel: regions that do not match but correspond in location; Forward chain: The forward chain of the genome sequence, where the gene coordinates increase in the clockwise direction; Reverse chain: The reverse chain of the genome sequence, where the gene coordinates increase counterclockwise.)

**Figure 5 ijms-25-09743-f005:**
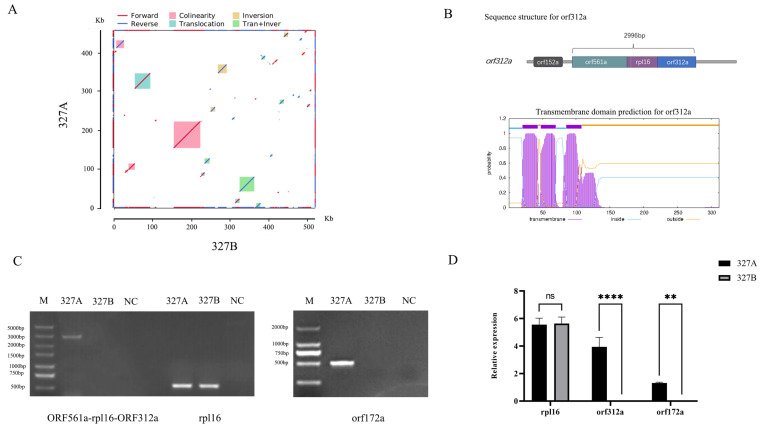
327A-specific ORF analysis and identification of CMS candidate genes. (**A**) collinearity comparison between 327A and 327B; (**B**) orf561a-rpl16-orf312a co-transcription chimera structure and corresponding transmembrane domain; (**C**) orf561a-rpl16-orf312a, *rpl16*, *orf172a* RT-PCR semi-quantitative analysis; (**D**) *rpl16*, *orf312a*, *orf172a* RT-QPCR relative expression analysis, **** *p* < 0.0001, ** *p* < 0.01.

**Figure 6 ijms-25-09743-f006:**
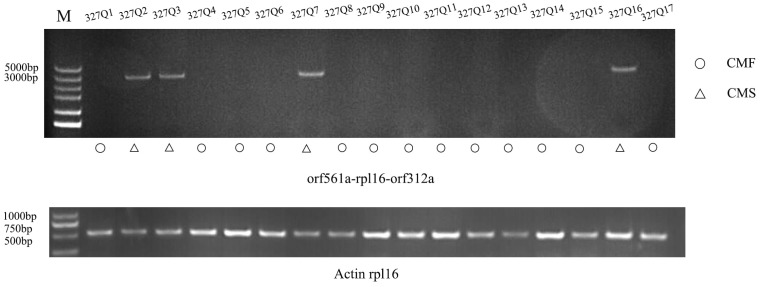
CMS markers were verified in 17 eggplant inbred lines.

## Data Availability

Data are contained within the article and Appendix A.

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
