# Peer review of "Comparative Analysis of the Mitochondrial Genome of Eggplant (Solanum melongena L.) to Identify Cytoplasmic Male Sterility Candidate Genes"

_ijms, 2024, doi:10.3390/ijms25179743_

Round 1

Reviewer 1 Report

Comments and Suggestions for Authors

 The study reports the discovery of a cytoplasmic male sterility (CMS) line, 327A, derived from the hybridization between cultivated and wild eggplants, where the anther's tapetum layer vacuolized during meiosis, causing abortion. The mitochondrial genomes of 327A and its maintainer line 327B were assembled de novo, revealing 15 unique ORFs in 327A, with orf312a and orf172a, two 327A-specific ORFs with transmembrane domains, strongly expressed in sterile anthers, making them strong candidate genes for CMS, and the authors have aimed at developing a functional molecular marker, CMS312, targeting the future theoretical basis for eggplant CMS three-line molecular breeding. Overall, the study covers some interesting aspects of eggplant genetics, however, in my opinion, the manuscript requires significant improvements, due to several major flaws listed below.

-          Latin names of species and names of genes should be written in italics.

-          Keywords should be arranged alphabetically and not repeat words from the title.

-          The authors should use the term ‘cultivar’ (product of breeding) instead of ‘variety’ (of wild origin).

-          Line 43: style.

-          Correct the reference style (line 52).

-          Details on the producers of key chemicals, equipment, and software should be used (including the name, city, state and country).

-          Space bar errors require correction (e.g. line 170).

-          Line 149: grammar.

-          Do not repeat Materials and methods in the Results section.

-          The Discussion is quite shallow and practically lacks references.

-          Reference list does not follow the MDPI format and lacks DOI numbers.

Comments on the Quality of English Language

The English form is generally understandable, although grammar and style require correction.

Reviewer 2 Report

Comments and Suggestions for Authors

Deng et al. have significantly contributed to our understanding of the molecular mechanisms underlying cytoplasmic male sterility (CMS) in eggplant by conducting a comparative analysis of the mitochondrial genomes between a CMS line and its maintainer line. Identifying specific candidate genes (orf312a and orf172a) associated with CMS and developing a molecular marker represents a critical advancement in improving the efficiency of hybrid seed production in eggplant.

The research is methodologically robust, employing advanced genomic techniques, including next-generation sequencing, and provides valuable resources for future investigations into the genetic basis of CMS.

However, I have several comments and suggestions for improvement:

Data Availability: The manuscript lacks a data availability statement, and the authors do not provide information on where the data can be accessed. Ensuring transparency and reproducibility is essential, and the authors should include a statement regarding the availability of the datasets used in the study.

Introduction: The introduction should offer a more comprehensive overview of the genetic basis of sterility, providing the reader with a clearer context and background on the subject.

Citation for Mitochondrial Genome Characteristics (Lines 34-35): The statement "The plant mitochondrial genome often contains many repeated sequences, and the size of the mitochondrial genome varies widely between species, with a range of 200–2400 kb" should be appropriately cited. Please provide a reference for this information.

Clarification on Methodology (Lines 104-105): The manuscript mentions that "Candidate sequences with excellent coverage depth and long assembly lengths were compared to the NT collection for confirmation." The authors should elaborate on the specific methodology used for this comparison, providing detailed information to enhance the clarity and reproducibility of the study.

Gene Prediction and Reference Genome (Lines 108-112): The authors should provide a detailed explanation of how the genes were predicted, including the specific criteria and tools used. Additionally, they should clarify which reference genome was utilized for this analysis.

Tool Versions: It is recommended that the authors verify the tools used in the study and include the version numbers, such as for AUGUSTUS, to ensure methodological transparency.

Figure 3: The mitochondrial assembly maps presented in Figure 3 should be enlarged for better visualization. Additionally, the caption should be expanded to describe the elements shown, such as the meaning of the color blocks, the significance of the blocks outside the circle, and the purpose of the inner gray-colored circles. The authors should also indicate which software was used to generate these maps.

Figure 6: The resolution of Figure 6 should be increased to enhance its readability and ensure that all details are clearly visible.

Conclusions: The conclusions section should be expanded to discuss the study's limitations, such as the absence of direct functional validation, and to outline potential directions for future research. This will provide a more balanced perspective on the study's contributions and areas for further exploration.

Reviewer 3 Report

Comments and Suggestions for Authors

The article is well written and provides interesting information regarding cytoplasmic male sterility (CMS) in eggplant. Furthermore it introduces the “discovery” a functional molecular marker associated to CMS. As such I have only a minor observation, concerning the Discussion section, that, given the good level of novelty of the presented results, -in my opinion- is rather poor. For instance, the possible future implications of the findings might be considered. Moreover, a more in-depth description of the two lines (327A and 327B) would certainly be interesting (e.g are they suitable for a commercial exploitation?)

Minor checks:

Abstract: Line 12, “In this study, we reported a …” , I suggest to use “we report”

Line 137, “We extracted the total RND” please clarify

Supplementary Figure S1 “length of reapeats statistics for 327A and 327B”, please correct into “repeats”

Round 2

Reviewer 1 Report

Comments and Suggestions for Authors

The authors followed all of my suggestions. The manuscript is now suitable for publication.

Comments on the Quality of English Language

The manuscript will benefit from the editorial language correction during publication process.

Reviewer 2 Report

Comments and Suggestions for Authors

I am happy with the updated version of the manuscript and thank the authors for their efforts.

Comments on the Quality of English Language

I think the manuscript needs to be entirely proofread.